# Efficacy of Sound Therapy for Tinnitus Using an Enriched Acoustic Environment with Hearing-Loss Matched Broadband Noise

**DOI:** 10.3390/brainsci12010082

**Published:** 2022-01-06

**Authors:** María Cuesta, Christiam Garzón, Pedro Cobo

**Affiliations:** 1Institute for Physical and Information Technologies (ITEFI), Spanish National Research Council (CSIC), 28006 Madrid, Spain; m.cuesta@csic.es; 2Facultad de Ingenierías y Ciencias Agropecuarias, University of The Americas, Avenue Granados y Colimes, Quito E12-41, Ecuador; christiam.garzon@udla.edu.ec; 3Instrumentation and Applied Acoustics Research Group, Polytechnic University of Madrid (UPM), 28031 Madrid, Spain

**Keywords:** tinnitus, hearing loss, sound therapy, enriched acoustic environment

## Abstract

Background: Tinnitus is a rather heterogeneous chronic condition/disorder which is difficult to treat. Some tinnitus treatments combine sound therapy with counselling. The main goal of this study is to report the efficacy of a customized sound therapy combined with counselling on a cohort of 83 tinnitus patients. Methods: 119 tinnitus subjects, recruited between January 2018 and June 2021, were subjected to a treatment consisting of a combination of an initial counselling session and four-month sound therapy. The sound stimulus was a personalized broadband noise colored by the audiometry of the subjects. These stimuli were given to the patients in mp3 format to be heard 1 h per day over 4 months. The tinnitus severity of the patients was evaluated monthly through the validated Spanish version of the Tinnitus Handicap Inventory. Results: Of the patients, 30% (36 of 119) withdrew from the treatment before finishing, and 96% (80 of 83) of the subjects completing the therapy attained some relief after 4 months. The overall average THI decrease of these 80 participants was 23. However, when the THI was analyzed by severity scales, it was found that patients with initial mild, moderate, severe and catastrophic handicap had an average THI decrease of 14, 20, 31 and 42 points, respectively. Thus, the average THI decrease depended on the baseline severity scale of patients. Conclusions: Consequently, the proposed treatment was demonstrated to be effective in providing clinically relevant relief in tinnitus distress patients in just 4 months.

## 1. Introduction

Tinnitus has been defined as the conscious perception of sound without physical sound sources external or internal to the body [1]. Although epidemiological studies account for a variable prevalence of tinnitus, it can affect roughly 10–15% of the adult population [2,3], and have a severe impact on the regular life of about 0.5–2% of adults by annoyance, irritation, disturbing sleep patterns, and producing panic, anxiety, and/or depression [3,4,5,6].

Most cases of persistent tinnitus are associated with hearing losses (HL). Eggermont [7] reports an approximately cubic-root dependence between the prevalence of tinnitus and the prevalence of HL, with a higher prevalence for tinnitus at ages < 45 years and a progressively reduced prevalence of tinnitus for older ages compared to the prevalence of HL. When subjects with tinnitus and HL are asked to match the pitch of their tinnitus, some degree of correlation is found between certain characteristics of the HL curve and the tinnitus pitch [8,9,10]. Cuesta and Cobo [11] found that the frequency at which hearing loss reaches 50 dB is the best audiometric correlate of tinnitus pitch.

Although HL is considered the main risk factor to developing tinnitus, not all subjects with HL suffer of tinnitus and not all tinnitus patients have HL. Recent research reported that a percentage of tinnitus patients (between one-tenth and one-third) have normal hearing (HL less than 25 dB), at least between 125 and 8000 Hz [3].

Tinnitus perception seems to arise as an aberrant plastic mechanism of the subcortical and cortical auditory system to compensate for a distorted peripheral auditory input. Eggermont [12] suggested three main mechanisms, namely, neural synchrony (hypersynchrony), reorganization of the tonotopic map, and increased spontaneous firing rates (hyperactivity) as the neural correlates of tinnitus in the auditory system.

Tinnitus is known to be a heterogeneous disorder in at least four dimensions [13]: perception, associated distress, multiple factors of risk and related comorbidities, and large variability in the response to treatments. This heterogeneity makes tinnitus a disorder difficult to treat. In fact, there are few treatments which are aimed at improving the impact rather than offering a cure of tinnitus [3]. Although there are several options for tinnitus treatment, including sound-based and psychological interventions, magnetic and electrical stimulation, as well as other therapies, internationally extended practical guidelines recommend counselling combined with sound therapy. Tinnitus Retraining Therapy (TRT) uses random noise in the audio frequency range as a sound stimulus [14,15,16,17]. Two features of sound therapy, namely, the dosage (number of hours per day) and the duration of treatment are quite variable in the studies. The positive effects of TRT can be perceived within several months, with moderate improvement in roughly 6 months, and high control of distress in approximately 12 months [14,18]. Cognitive behavioural therapy (CBT), an inclusive term that features and combines numerous psychological interventions that have been developed and evolved from cognitive and behavioural therapies respectively, is another therapy that has had some efficacy in tinnitus treatment [19]. CBT for tinnitus aims primarily to reduce the impact of tinnitus on quality of life, rather than directly change the perceived loudness.

In a recent study [20], 25 tinnitus patients with unique counselling sessions applied at baseline were exposed to one hour per day sound therapy over four months. The sound stimulus consisted of a personalized broadband sound with the spectrum filtered by the hearing loss (HL) curves of the patients. After these four months, 88% of patients obtained a clinically relevant relief of their tinnitus, as assessed by a mean reduction of 29 points in their Tinnitus Handicap Inventory (THI) [20]^.^ In this study, only patients with THI ≥ 32 were included. However, a decrease of 20 points in the THI score is considered as clinically relevant [14]. Therefore, the main purpose of this article is to provide updated results of the preceding study [20], including a larger number of tinnitus patients (83) with lower threshold baseline distress (THI ≥ 20). Furthermore, an analysis of the patient’s handicap improvement was carried out by severity scales.

## 2. Materials and Methods

### 2.1. Participants and Measurements

The study was approved by the CSIC’s Research Bioethics Subcommittee and was carried out in accordance with the General Data Protection Regulation (EU Regulation 2016/679). Informed consent was signed by all participants at the start of the recruitment. A total of 133 tinnitus volunteers (see Figure 1) were initially enrolled, between January 2018 and June 2021, through Spanish Tinnitus Associations and Audiological Clinics.

The criteria to be included were subjects older than 18 and younger than 75 years with initial THI ≥ 20 and non-pulsatile tinnitus. Patients with severe Meniere syndrome, hydrocephaly, and severe psychiatric distress were excluded. This resulted in 119 included and 14 excluded subjects for treatment. Although this study required only a short period (4 months), 36 subjects quitted the treatment before finishing. This left 83 treated patients (see Figure 1), and 80 of these subjects achieved some tinnitus relief after four months of treatment. The baseline characteristics and demographics of patients who completed the treatment are summarized in Table 1. Averages for age and tinnitus duration are expressed as mean with standard deviation (SD) values. Their (mean, SD ) age was (51.6, 9.1) years. Their (mean, SD) tinnitus duration was (6.3, 8.8) years. There were 52 male participants with (mean, SD) age of (52.9, 8.9) years and (mean, SD) tinnitus duration of (7.2, 9.3) years, and 31 female subjects with (mean, SD) age (49.4, 9.6) years and (mean, SD) tinnitus duration (5.0, 7.8) years.

All subjects underwent standard HL measurements and tinnitus assessment as part of a clinical protocol which varied over the time related to the beginning of the COVID-19 lockdown (14th March 2020 in Madrid, Spain). In the pre-COVID-19 period, the participants underwent standard HL measurements of both ears at eleven frequencies (125, 250, 500, 750, 1000, 1500, 2000, 3000, 4000, 6000, and 8000) Hz in our laboratory using a Clinic Audiometer GSI 60. Post-COVID-19 participants were required to provide the audiometry of both ears measured in other exterior Audiological Clinic. Figure 2 shows the average HL curves of participants. Shaded areas around the mean curves display the 95% confidence (1.96 x SD/sqrt(N)) intervals. On average, tinnitus participants showed high frequency hearing losses, slightly greater at the left ear.

Average audiometric thresholds (*AAT*) were first calculated for the left and right ears. *AAT* is defined as
(1)$$AAT=\frac{1}{Nf}\sum_{125}^{8000}HL\left(f_{i}\right),$$
where *HL(f_i_)* represents the HL values at each frequency and *Nf* is the number of frequencies (11 in this case).

Additionally, average audiometric thresholds at low frequencies *(AAT_LF_)* and high frequencies (*AAT_HF_*) were calculated as the average for the seven frequencies from 125 to 2000 Hz, and four frequencies from 3000 to 8000 Hz, respectively
(2)$$AAT_{LF}=\frac{1}{7}\sum_{125}^{2000}HL\left(f_{i}\right),$$
(3)$$AAT_{HF}=\frac{1}{4}\sum_{3000}^{8000}HL\left(f_{i}\right).$$

The hearing condition of the participants was categorized according to the following criterion [20]:

If any(*HL(f_i_)*) ≥ 40 dB

or *AAT* ≥ 30 dB

or |*AAT_HF_ − AAT_LF_*| ≥ 17 dB

the subject is Hearing Impaired (HI)

else, the subject has Normal Hearing (NH).

Table 2 summarizes the average audiometric thresholds (*AAT*) for the participants gathered by subgroups. It is noticeable that 27 % (22/83) of the tinnitus participants had normal hearing. 61/83 participants (73%), on the other hand, had high-frequency hearing losses. Figure 3 and Figure 4 show the average HL curves for the NH and HI subgroups of tinnitus participants, respectively.

To assess their tinnitus features, the participants were asked to fill out a clinical evaluation sheet questioning about temporal (stable, fluctuant, intermittent), spectral (number of sounds, pitch), spatial (left ear, right ear, bilateral or central) aspects of their tinnitus, as well as their tinnitus onset. In addition, the patients were asked about their clinic history and tinnitus severity, through a Spanish version of the Tinnitus Handicap Inventory (THI) [21]. The clinical history included information about the descriptive characteristics and probable onset of their tinnitus, significant comorbidities, and earlier treatments (other sound therapies, acupuncture, pharmacology, physical therapy, cognitive behavioral therapy, and etcetera).

A custom-designed Graphical User Interface (GUI) was used to match the type and pitch of the tinnitus. This GUI, Figure 5, generated band-pass filtered random noise. The peak frequency and bandwidth of the band-pass filter at half amplitude determined the type of sound. Specifically, tones, ringing and hissing sounds were provided by bands-pass lesser than 0.1%, between 0.1 and 10%, and greater than 10%, respectively.

The tinnitus pitch matching procedure varied as a consequence of the COVID-19 lockdown. Pre-COVID-19 participants searched for the band-pass filtered noise closer to their tinnitus at the computer in our laboratory. First, they were acquainted with the effect of peak frequency and bandwidth of the sound. Then, they were requested to identify roughly the sound of their tinnitus (tone, ringing or hissing). Next, the effect of varying the peak frequency and bandwidth on the sounds were trained. Lastly, each patient was requested to match as closely as possible its tinnitus sound to that generated by the GUI. Some participants referred several types of sounds. In these cases they were required to match all these sounds by running successively the GUI. For post-COVID-19 participants, this procedure was successfully carried out by videoconference. The assessed tinnitus features of the 83 participants are summarized in Table 3.

The predominant tinnitus onset of the cohort was associated with psychiatric disorder (40%), followed by hearing loss (32%), overexposure to noise (22%), ear surgery (7%), tube dysfunction (7%), idiopathic (6%), and others (head trauma, ototoxicity, otitis, rhinitis and sinusitis, barotrauma, and cervical troubles). Taking into account that some patients could assign their tinnitus to several causes, the sum of onset percentages was higher than 100.

### 2.2. Tinnitus Treatment

Tinnitus patients underwent a combination of counseling and sound therapy. According to the neurophysiological model of Jastreboff [14], the harmful emotional effects of tinnitus can be greatly diminished by turning off the aberrant exacerbated loop between the limbic and autonomic nervous systems with the auditory system. When the patient is able to understand the fundamentals and mechanisms causing its perception, he/she can achieve desensitization and habituation to tinnitus. This is the main goal of counselling. In this study, counselling was provided over a unique session of roughly 60 min using a standardized PowerPoint presentation. Initially, an explanation was provided of the auditory system, paying special attention to the neural part. Then, a description of the main tinnitus mechanisms was given, i.e., hypersyncrhony, hyperactivity, and reorganization of the tonotopic map as plastic compensation reactions to deafferentation to the auditory system [12]. The comprehension of the neural connections between the auditory and limbic systems helped to clarify the strong emotional reactions to tinnitus, such as annoyance, irritability, stress, anxiety, panic, and depression.

Once the tinnitus mechanisms were understood, subjects could be easily introduced to the fundamentals of the neurophysiological model of Jastreboff [14], and they were instructed in how brain plasticity can help to decrease these strong emotional reactions by blocking the loop between the auditory and the limbic and autonomic nervous systems. However, this is a slow process requiring several months, so that tinnitus patients must have adherence to the treatment.

The main goal of sound therapy is to vary auditory processing at the subconscious level to stimulate habituation to tinnitus perception [15]. Numerous sound stimuli have been proposed for tinnitus therapy in the past decades [22,23]. According to Jastreboff and Jastreboff, any type of sound which does not annoy, create discomfort or damage hearing, is better than silence [18]. However, Schaette and Kempter [24] used a computer model of tinnitus-related hyperactivity to show that a personalized sound consisting of broadband noise filtered by the audiometry of the patient should reduce tinnitus optimally. Such a customized broadband sound stimulus was generated by using a custom-designed GUI (Figure 6), consisting of an enriched acoustic environment (EAE) of filtered broadband noise [25]. As the HL curves of both ears are used for filtering the random noise, these stimuli are always stereo, regardless of the ear location of the tinnitus.

These customized EAE stimuli were given to the participants in mp3 format with instructions to listen to them for one hour each day for four months, using any audio player device (mp3 player, smartphone, or computer) connected to high quality headphones. Furthermore, they were trained in adjusting the stimuli volume to a level just below their tinnitus percept (the mixing point) [14].

A validated Spanish version of the THI [21,26], designed to measure the distress, handicap and psychosocial consequences of tinnitus, was used to assess the outcome of the proposed treatment. During the counselling session, participants were instructed to fill in this questionnaire. According to the severity scale of tinnitus proposed by McCombe et al. [27], 40% of participants had a mild handicap (18 ≤ THI ≤ 36), 25% suffered a moderate handicap (38 ≤ THI ≤ 56), 17.5% were concerned by a severe handicap (58 ≤ THI ≤ 76), and 17.5% experienced catastrophic handicap (THI ≥ 78). After assessing initial values of THI, and until the end of the treatment, participants were requested monthly by email to fill in new THI questionnaires. A decrease of 20 points in the THI score is considered to be clinically relevant [14].

## 3. Results

Among the 83 participants completing the study, 80 (96.4%) benefited from the treatment while three failed to get any relief (their THI score even increased by several points). The average THI change over the 4 months of treatment for the 80 successful participants is presented in Figure 7. The average decrease of THI score for these 80 participants was 23 points, which is clinically relevant [14].

Notice that this overall THI reduction (23 points) is slightly lower than that reported by Cuesta and Cobo [20], which was 29 points for 22 patients. The main reason is that the current study included patients with THI ≥ 20, while the previous study excluded patients with THI ≤ 32. It could be expected that the greater the initial THI, the more reduction potential there should have been. This is illustrated in Figure 8, where the tinnitus distress reduction analysis is carried out by severity scales. Table 4 summarizes the result of this analysis. We therefore confirm that the final THI reduction of the treatment increased with the baseline value of THI.

For the mild and moderate scales, the mean THI score was reduced by 14 and 20 points, respectively. However, for the severe and catastrophic scales, much more reductions were obtained (31 and 42 points, respectively).

Figure 9 shows a bar diagram of the THI reduction by scales. One-way ANOVA tests showed that differences between means before and after treatment are statistically different at *p* < 0.001. All the participants passed to lower distress scales. On average, the 32 patients on the mild scale moved to the slight or no handicap scale (one grade fall), the 20 patients with moderate handicap came down one grade to mild tinnitus, the 14 subjects with severe tinnitus descended two grades to mild handicap, and the 14 individuals with catastrophic tinnitus moved down two grades to moderate tinnitus.

## 4. Discussion

Although this is a non-controlled, small-size observational study, the efficacy of TRT was earlier confirmed by randomized clinical trials [15,16,17,28]. The aim of this study was to assess the improvement of the results by using personalized treatment based on matching the broadband noise frequently used in TRT with the hearing loss curves of patients. A control group was discarded due to ethical reasons [29]. Nevertheless, we are conscious that the study should be tested in a larger sample size to increase its statistical power.

Our categorization of HL subtypes considered hearing impaired (HI) and normal hearing (NH) subtypes [30]. The HI subgroup was the most prevalent HL subtype (73%), in our study, followed by NH (27%). The percentage of tinnitus patients with apparently normal hearing in this study (27%) was in between those of other authors. In Lindblad et al. [31], one third of tinnitus occurred without simultaneous HL. McFerran el al. [3], on the other hand, suggested that up to one in 10 tinnitus patients could have normal pure tone audiometry. Nevertheless, HL above 8 kHz or cochlear synaptopathy, two causes of auditory deafferentiation, can coexist with normal HL up to 8 kHz [32,33]. Cochlear synaptopathy consists of the selective loss of synaptic connections of the auditory nerve with high-threshold and low-spontaneous rate inner hear cells, which does not impair hearing thresholds. Since the loss of these connections can reach 40-50 % without elevating hearing thresholds, this hearing disorder is also named hidden hearing loss (HHL) [33].

Of the participants enrolled in this study, 70% (83 of 119) finished the treatment, while 30% (36 of 119) left it before concluding. Although we do not have an explanation for this high abandon ratio (subjects did not reply to our follow up email), some participants in this study had had negative experience with preceding treatments which turned them into rather impatient patients.

Of patients completing the treatment 96% (80 of 83) reached clinically relevant relief after 4 months. Differences in the therapy outcome between NH and HI patient subgroups were not found. It is remarkable that, as already noted by Jastreboff, tinnitus onset is not important for the success of the therapy [14]. This success ratio is higher than the 80% frequently declared for other TRT studies [14,16,28]. The THI decay (ΔTHI) of our treatment obtained in just 4 months is comparable to those mentioned by others studies with a longer duration. For instance, a 12 months TRT study by Jastreboff accounted for a ΔTHI of 23 points after the first 4 months [14]. Henry et al. reported a controlled clinical study to evaluate prospectively the clinical efficacy of tinnitus masking and TRT in military veterans having clinically significant tinnitus [15]. After the first 4 months they found a ΔTHI of 7 points.

Many other studies have used also sound stimuli different to simpler broadband noise in the audio frequency. Henry et al. [34] showed that the effectiveness of TRT can be improved by coloring the typical broadband noise. However, Barozzi et al. [35] found that there were no significant differences between nature and technical sounds in the management of patients suffering from tinnitus. Similar improvements were obtained in both groups at three and six months following the TRT treatment. Kim et al. [36], on the other hand, compared the effectiveness of three band filtered noises in the treatment of 38 tinnitus subjects over 9 weeks; specifically, narrowband noise TRT (nTRT), mixed band noise TRT (mTRT), and broadband noise TRT (bTRT). They found that, although the three sounds can provide relief in patients with annoying tinnitus, the bTRT group showed a higher success rate than other groups. We claim that the high success ratio of our study is related to the personalized sound stimulus used in the treatment, which is the optimal sound to reverse tinnitus according to Schaette and Kempter’s model [23].

Our results corroborate previous results on the dependency of the ΔTHI after TRT with the baseline distress of individuals. For instance, Oishi et al. [37] assessed the effect of TRT employing monaural noise generators on 95 patients with chronic tinnitus. They found that the ΔTHI of severely distressed patients, after 6 months of TRT, almost doubled the ΔTHI experienced by moderately distressed patients.

According to De Ridder et al. [38] “Tinnitus is the conscious awareness of a tonal or composite noise for which there is no identifiable corresponding external acoustic source, which becomes Tinnitus Disorder when associated with emotional distress, cognitive dysfunction, and/or autonomic arousal, leading to behavioural changes and functional disability”. It must be emphasized that this therapy should be considered in the area of treatment of tinnitus disorder, as it includes audiological (sound) and psychological (counselling) intervention.

This study has some limitations. First, high frequency hearing losses (>8000 Hz) of patients were not obtained, which could modify the percentage of subjects classified as having normal hearing (NH) [32]. Second, psychoacoustic characterization of tinnitus was not performed, which could be relevant to understanding individual responses to sound therapy. Third, this is an ongoing study, so that its statistical strength should increase as the sample size grows. While the pre-post treatment results were highly promising in comparison with other similar studies, we also noted a high attrition rate (36 out of 119 patients) as participation was on a completely voluntary nature. Taking into account the positive pilot results of the therapy [25], and compelled by ethical reasons, we were not able to implement a completely randomized and controlled clinical trial with a comparison group. If possible, future studies need to consider inclusion of a comparison control group.

## 5. Conclusions

The efficacy of conventional treatment of tinnitus (Tinnitus Retraining Therapy) can be improved by designing customized sound stimuli matched to the hearing loss curves of the patients (Enriched Acoustic Environment). The main benefit of this refinement is yielding a statistically significant and clinically relevant distress reduction in a shorter period of time. In this study, 80 of 83 tinnitus patients of heterogeneous onset achieved an average THI score reduction of 23 points in just four months of treatment. Furthermore, this reduction was dependent on the initial severity scale of the patients. In general, the greater the initial severity degree, the greater the final THI reduction.

## Figures and Tables

**Figure 1 brainsci-12-00082-f001:**
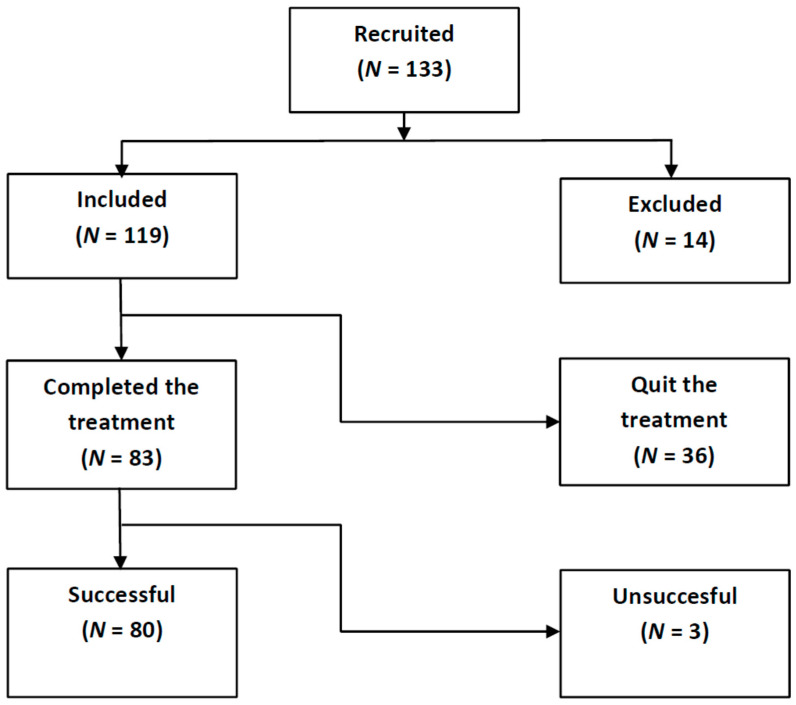
Flow diagram of participant enrollment, intervention and analysis in the current study.

**Figure 2 brainsci-12-00082-f002:**
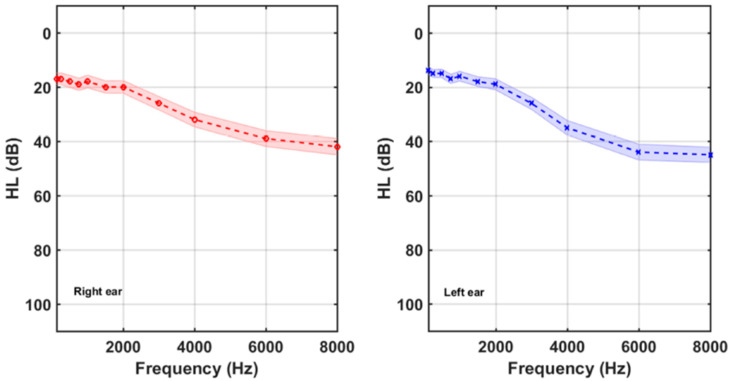
Right and left average audiograms of the 83 participants. Shaded areas around the mean curves display the 95% confidence intervals.

**Figure 3 brainsci-12-00082-f003:**
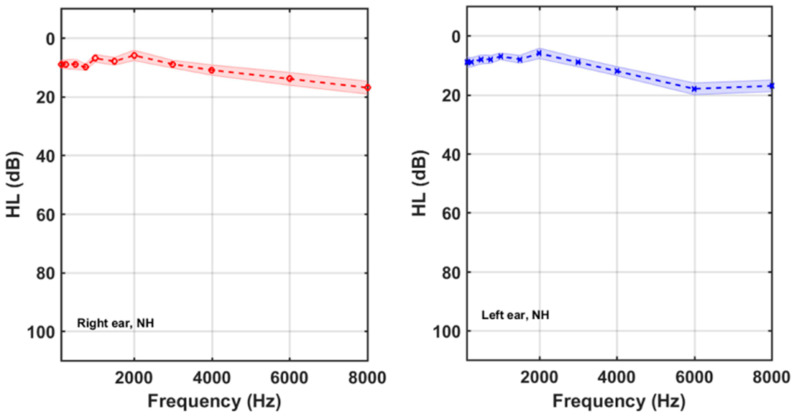
Right and left average audiograms of the 22 participants of the NH subgroup. Shaded areas around the mean curves display the 95% confidence intervals.

**Figure 4 brainsci-12-00082-f004:**
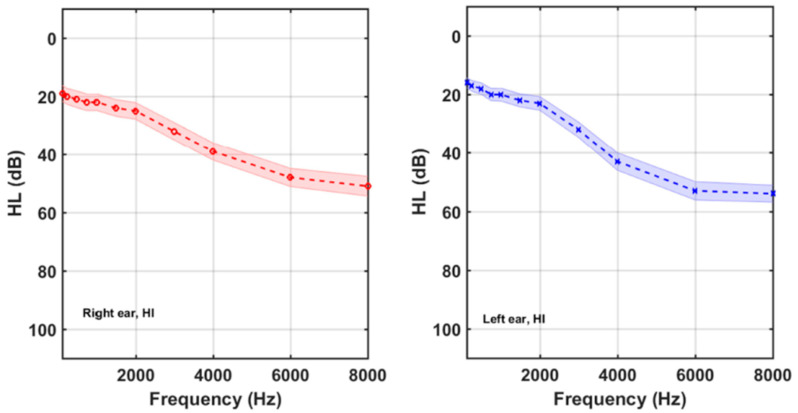
Right and left average audiograms of the 61 participants of the HI subgroup. Shaded areas around the mean curves display the 95% confidence intervals.

**Figure 5 brainsci-12-00082-f005:**
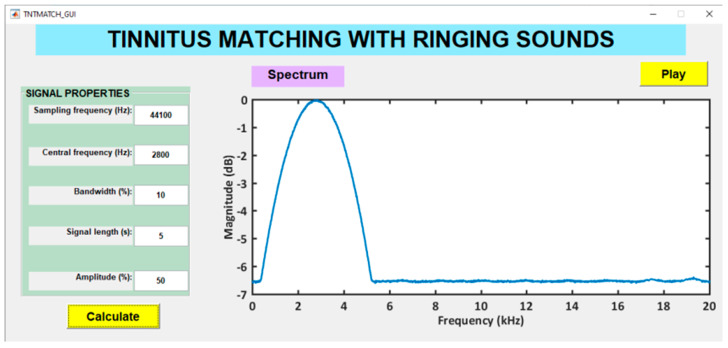
Custom-designed GUI for tinnitus pitch matching.

**Figure 6 brainsci-12-00082-f006:**
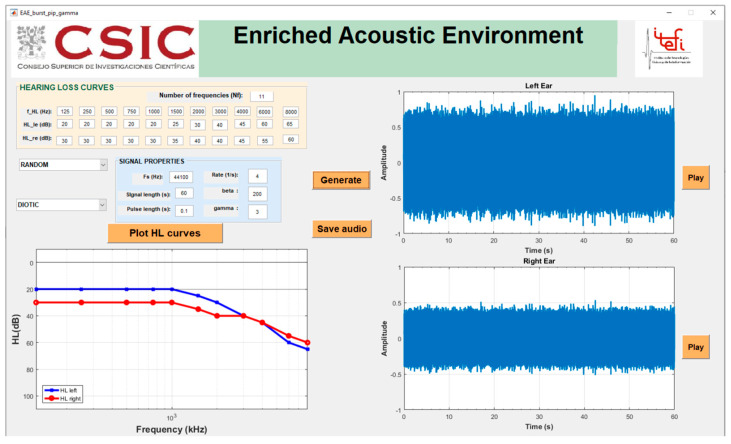
Custom-designed GUI for tinnitus sound therapy.

**Figure 7 brainsci-12-00082-f007:**
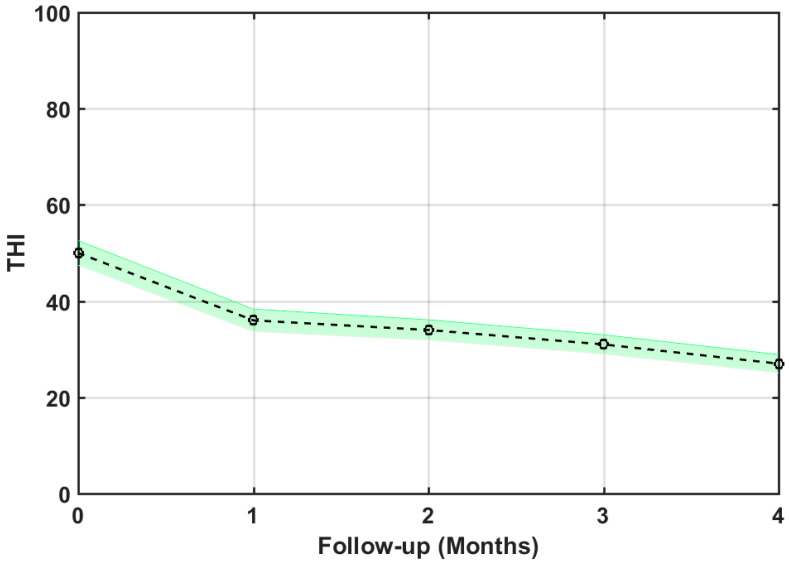
Average THI score change over time for the described treatment. Shaded areas around the mean THI display the 95% confidence intervals.

**Figure 8 brainsci-12-00082-f008:**
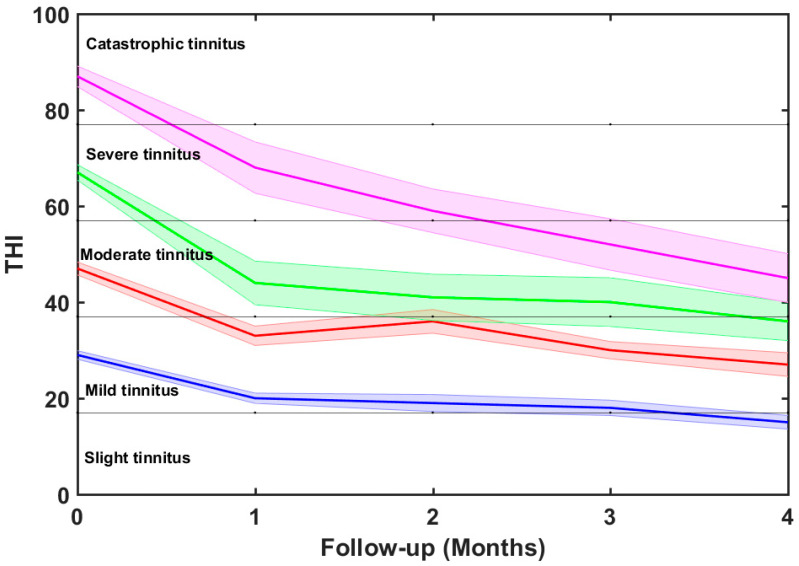
Average THI score change over time for the different distress groups. Shaded areas around the mean THI display the 95% confidence intervals.

**Figure 9 brainsci-12-00082-f009:**
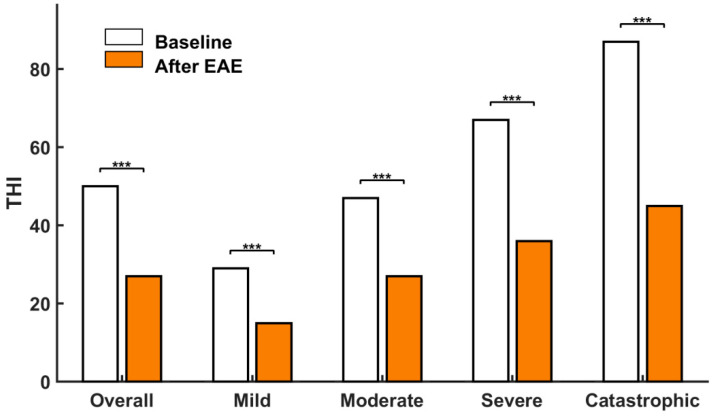
Average THI scores before and after treatment for the different distress groups. *** represents statistical significance at *p* < 0.001.

**Table 1 brainsci-12-00082-t001:** Baseline characteristics and demographics of patients who completed the treatment.

		Age (Years)		Tinnitus Duration (Years)	
Group	*N*	Mean	SD	Mean	SD
All	83	51.6	9.1	6.3	8.8
Male	52	52.9	8.9	7.2	9.3
Female	31	49.4	9.6	5.0	7.8

**Table 2 brainsci-12-00082-t002:** Average audiometric thresholds (*AAT*) of patients by subgroups.

		*AAT* (dB)				*AAT_LF_* (dB)				*AAT_HF_* (dB)			
		Left Ear		Right Ear		Left Ear		Right Ear		Left Ear		Right Ear	
Group	*N*	Mean	SD	Mean	SD	Mean	SD	Mean	SD	Mean	SD	Mean	SD
All	83	24	15	24	19	16	13	18	19	37	22	34	23
NH	22	11	5	10	5	9	6	9	6	15	6	13	7
HI	61	29	14	29	20	19	14	21	21	45	20	42	22

**Table 3 brainsci-12-00082-t003:** Tinnitus features of patients by subgroups.

		Tinnitus Location (%)			Tinnitus Sound (%)			Tinnitus Pitch (Hz)		Initial THI	
Group	*N*	Left	Right	Bilateral	Hissing	Ringing	Tonal	Mean	SD	Mean	SD
All	83	36	21	43	44	26	30	5213	3054	50	23
NH	22	43	14	43	38	43	19	4475	3870	52	23
HI	61	34	24	42	46	20	34	5479	2695	48	22

**Table 4 brainsci-12-00082-t004:** Average THI reduction by severity scales.

Severity Scale	*N*	THI_ini_	THI_fin_	ΔTHI
Mild	32	29	15	−14
Moderate	20	47	27	−20
Severe	14	67	36	−31
Catastrophic	14	87	45	−42
All	80	50	27	−23

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
