# Peer review of "Efficacy of Sound Therapy for Tinnitus Using an Enriched Acoustic Environment with Hearing-Loss Matched Broadband Noise"

_brainsci, 2022, doi:10.3390/brainsci12010082_

Round 1
Reviewer 1 Report
The submitted manuscript describes results of a therapy targeted at lowering tinnitus-induced distress in a sample of patients with and without hearing loss defined with pure tone audiometry between 125 and 8000 Hz.
Abstract:
“Tinnitus is a rather heterogeneous disorder which is difficult to treat.”
Would you please revise that sentence? Tinnitus is not a disorder but a symptom of various disorders.
“Usual tinnitus treatment practice combines sound therapy with counselling.”
Please revise. Therapy for tinnitus is firstly directed at the cause of tinnitus (hearing loss, cardiovascular conditions, diabetes, ect.).
“The sound stimulus was a personalized broadband noise colored by the audiometry of the subjects”
A sound colored by the audiometry??? Please revise.
The entire abstract needs to be rewritten; syntax, grammar, and punctuation errors must be corrected.
Hearing loss is only one of the diseases associated with tinnitus. In fact, in lines 168-169, the Authors state that “The predominant tinnitus etiology of the cohort was psychiatric disorder (40%), followed by hearing loss (32%), overexposure to noise (22%), ear surgery (7%), tube dysfunction (7%), idiopathic (6%), and other…”, which is in contrast to the statements in the Introduction. However, the etiologies of tinnitus provided by Authors need clarification. Which “psychiatric disorders” caused tinnitus? Do Authors have evidence about psychiatric disorders CAUSING tinnitus? Or were the psychiatric disorders comorbid conditions accompanying tinnitus? Second, why separate the patients with hearing loss from those who were “overexposed to noise”? What were the reasons for ear surgery? Moreover, the numbers do not add up. In the part where the PTA results are presented, the authors state that 27% of patients had normal hearing thresholds, which means that 73% of patients had hearing loss. That is in contrast with lines 168-169. Please revise.
The TRT is a well described method (see Scherer, R.W., Formby, C., Gold, S. et al. The Tinnitus Retraining Therapy Trial (TRTT): study protocol for a randomized controlled trial. Trials 15, 396 (2014). https://doi.org/10.1186/1745-6215-15-396) and the name “TRT” should only be used when the protocol is fully followed. Based on the methodological description, it is apparent that the Authors modified the TRT method. Therefore, the modifications should be clearly stated, and the name “TRT” should not be used.
Although in the Methods, the Authors went into the trouble of sample stratification into with and without hearing loss, in the Results, the Authors do not demonstrate therapy outcome based on the patients’ hearing loss status. Why not?
Author Response
Please see the attachement

Reviewer 2 Report
The subject of this paper was a customized sound therapy - extension of the research conducted by the authors in the past [reference 19] on a larger group of patients suffering from tinnitus to provide the recent findings. Strengths: clear and practical approach, consistent message regarding the search for new therapies for the tinnitus treatment. Slight weakness: too short observation time for severe and catastrophic cases.
My comments:
94: Fig. 1 : Why fourteen people were excluded initially?
140: How was this clinical history taken into account? Because if they have undergone similar therapies before, the changes in this experiment may be slight - the drop of delta THI shown in Fig. 8 is decreasing in time.
206: in Fig. 6 – What do the beta and gamma parameters mean?
262: Maybe not only a larger sample size: Was the tinnitus also monitored for 4 months - how did its parameters change? - maybe then you will get something like feedback from patients (decribed in lines 159-164) and greater effectiveness of treatment for severe cases (see Fig. 9) by providing updated mp3 files during the experiment?
318 – 319: I think that the final THI reduction for the catastrophic, severe, and moderate tinnitus was not reached after 4 months (see Fig. 8). Shouldn't the experiment time be varied depending on the noise level?
Reviewer 3 Report
This is a non-randomized, non-controlled clinical trial that tries to measure the effect of counselling and sound therapy in a group of 119 tinnitus patients.
Although this is a pre/post intervention study with a medical device to provide sound therapy, an Ethical approval obtained from a Clinical Trial Ethical Review Board is needed and it is not clear if the authors have obtained this approval. The protocol and registry number for this trial should be included.
In general, the paper is clearly written, and the conclusions are supported by the findings.
Introduction
This is a short and focused introduction to the topic of sound therapy for tinnitus.
I suggest including CBT as an additional therapy that has proven some efficacy in RCT.
Methods
Description of the methods is enough including inclusion and exclusion criteria. However, the selection of participant raises some concerns. Tinnitus duration could be a relevant variable to investigate the effect of sound therapy and probably the authors should have analyzed their data according to tinnitus duration.
Another concern is the high number of drop-outs (36/119) which may indeed affect the final outcome.
Although a control group is not possible for ethical reason, another comparison therapy or different sound therapy schedule will probably have resulted in more robust conclusions
Figure 9 is not very friendly for color blindness people, please use the yellow/magenta combination recommended for main journals.
Discussion
At some point in the discussion the authors should differentiate between tinnitus and tinnitus disorder that may define which individuals need healthcare including audiological or psychological intervention. This could be estimated with THI scores.
A study limitation section with the weakness and strengths of the study should be added at the end of the discussion.
Questions
- Did the authors obtain high frequency thresholds (> 8000Hz)? This is relevant for the normal hearing group?
- Did the authors perform a psychoacoustic characterization of tinnitus? MML, residual inhibition?...these variables may be relevant to understand individual responses to sound therapy and residual inhibition is found in around 30% of tinnitus patients.
Suggested references
Fuller T, Cima R, Langguth B, Mazurek B, Vlaeyen JW, Hoare DJ. Cognitive behavioural therapy for tinnitus. Cochrane Database Syst Rev. 2020 Jan 8;1(1):CD012614. doi: 10.1002/14651858.CD012614.pub2. PMID: 31912887; PMCID: PMC6956618.
De Ridder D, Schlee W, Vanneste S, Londero A, Weisz N, Kleinjung T, Shekhawat GS, Elgoyhen AB, Song JJ, Andersson G, Adhia D, de Azevedo AA, Baguley DM, Biesinger E, Binetti AC, Del Bo L, Cederroth CR, Cima R, Eggermont JJ, Figueiredo R, Fuller TE, Gallus S, Gilles A, Hall DA, Van de Heyning P, Hoare DJ, Khedr EM, Kikidis D, Kleinstaeuber M, Kreuzer PM, Lai JT, Lainez JM, Landgrebe M, Li LP, Lim HH, Liu TC,
Lopez-Escamez JA, Mazurek B, Moller AR, Neff P, Pantev C, Park SN, Piccirillo JF, Poeppl TB, Rauschecker JP, Salvi R, Sanchez TG, Schecklmann M, Schiller A, Searchfield GD, Tyler R, Vielsmeier V, Vlaeyen JWS, Zhang J, Zheng Y, de Nora M, Langguth B. Tinnitus and tinnitus disorder: Theoretical and operational definitions (an international multidisciplinary proposal). Prog Brain Res. 2021;260:1-25. doi: 10.1016/bs.pbr.2020.12.002. Epub 2021 Feb 1. PMID: 33637213.
Round 2
Reviewer 1 Report
Apart from one remark, the authors ignored all my suggestions for improvement. The main job of any reviewer is to make a given manuscript a bit better, easier to read, and of course, to clarify all technical and methodological discrepancies.
1) Abstract: “Tinnitus is a rather heterogeneous disorder which is difficult to treat.” Would you please revise that sentence? Tinnitus is not a disorder but a symptom of various disorders.
Response: There is an insightful discussion of this subject in one of the papers recommended by Reviewer #3 (De Ridder et al., Tinnitus and tinnitus disorder: Theoretical and operational definitions (an international multidisciplinary proposal), Prog Brain Res. 2021; 260: 1-25). According to these authors (page 12, first paragraph) “tinnitus disorder is the conscious awareness of a tonal or composite noise for which there is no identifiable corresponding external acoustic source, with associated suffering.”
They also discuss tinnitus in the domain of somatic symptom disorders (SSD). In this context, tinnitus severity could be graded according to the number of associated symptoms (it seems, therefore, that tinnitus disorder is a group of several symptoms). Thus, we keep the sentence “Tinnitus is a rather heterogeneous disorder which is difficult to treat” unchanged in the abstract.
The paper which authors refer to starts with a definition of tinnitus, which is “Tinnitus is a clinical symptom associated with changes in auditory and other brain systems (…)”.
In that paper, the authors proposed to introduce the following terms:
The definition of tinnitus: Tinnitus is the conscious awareness of a tonal or composite noise for which there is no identifiable corresponding external sound source.
The definition of tinnitus disorder: Tinnitus Disorder is the conscious awareness of a tonal or composite noise for which there is no identifiable corresponding external sound source, associated with emotional and/or cognitive dysfunction, and/or autonomic arousal, leading to behavioral changes and functional disability.
Since the Authors have not performed their research on tinnitus disorder, meaning they did not study the consequence of tinnitus in form of associated cognitive dysfunctions, autonomic arousal, etc., as defined above by the recently proposed definition, but rather were occupied with tinnitus from the audiological side, therefore the use of the term “tinnitus disorder” in this case is not substantiated. Please revise your definition and the terms used.
2) Abstract: “Usual tinnitus treatment practice combines sound therapy with counselling.” Please revise. Therapy for tinnitus is firstly directed at the cause of tinnitus (hearing loss, cardiovascular conditions, diabetes, ect.).
Response: We have changed “Usual tinnitus treatment practice combines sound therapy with counselling” for “Some tinnitus treatments combines sound therapy with counselling”. Furthermore, we have modified the next sentence “The main goal of this study is to report the efficacy of a customized sound therapy on a cohort of 83 tinnitus patients” for “The main goal of this study is to report the efficacy of a customized sound therapy combined with counselling on a cohort of 83 tinnitus patients”
3) Abstract: “The sound stimulus was a personalized broadband noise colored by the audiometry of the subjects” A sound colored by the audiometry??? Please revise.
Response: “Colored sound” or “colored noise” is a term very often used in audio engineering. For instance, a white (broadband) noise, by analogy with white light, is a noise with a flat power spectrum. However, pink noise is a noise which power spectrum falls off at 10 dB/decade (so, richer in low frequencies), and blue noise is a noise which power spectrum grows up at 3 dB/decade (so, richer in high frequencies). In our therapy, the sound stimulus is obtained by filtering a broadband (white) noise by the HL curves (audiometry) of the patient. This means exactly that the power spectrum of the white noise is colored by such curves.
The explanation is appreciated. However, the Authors submitted their work to Brain Sciences and not to an audio-engineering journal and should write to a Brain Sciences reader understandably.
4) Abstract: The entire abstract needs to be rewritten; syntax, grammar, and punctuation errors must be corrected.
Response: The entire abstract has been revised for syntax, grammar and punctuation mistakes.
5) Hearing loss is only one of the diseases associated with tinnitus. In fact, in lines 168-169, the Authors state that “The predominant tinnitus etiology of the cohort was psychiatric disorder (40%), followed by hearing loss (32%), overexposure to noise (22%), ear surgery (7%), tube dysfunction (7%), idiopathic (6%), and other…”, which is in contrast to the statements in the Introduction.
Response: We are not indeed aware which sentence in the Introduction is in contrast with tinnitus etiology. Please, take into account that a “state of the art” review is carried out in this Section, where sentences about tinnitus are attributed to their legitimate authors.
In the introduction, the Authors state that: “Most cases of persistent tinnitus are associated with hearing losses (HL). Eggermont [7] reports an approximately cubic-root dependence between the prevalence of tinnitus and the prevalence of HL, with a higher prevalence for tinnitus at ages < 45 years and a progressively reduced prevalence of tinnitus for older ages compared to the prevalence of HL.”
My understanding is that the Authors agree with that statement. Therefore, writing later that tinnitus etiology was in 32% the hearing loss is inconsistent.
6) However, the etiologies of tinnitus provided by Authors need clarification. Which “psychiatric disorders” caused tinnitus? Do Authors have evidence about psychiatric disorders CAUSING tinnitus? Or were the psychiatric disorders comorbid conditions accompanying tinnitus? Second, why separate the patients with hearing loss from those who were “overexposed to noise”? What were the reasons for ear surgery? Moreover, the numbers do not add up. In the part where the PTA results are presented, the authors state that 27% of patients had normal hearing thresholds, which means that 73% of patients had hearing loss. That is in contrast with lines 168-169. Please revise.
Response: First: We have not evidence of tinnitus caused by “psychiatric disorders”. We classified the etiology of the patients based on their clinical history. Many of them were diagnosed with psychiatric disorders (depression, anxiety disorder, insomnia, somatoform disorder) by their doctors. Thus, following the guidelines of TRI (2010) (see the following Figure) we assigned these patients to this etiology.
Here is the definition of etiology in medicine:
“the cause, set of causes, or manner of causation of a disease or condition.“
The Authors still write that “The predominant tinnitus etiology of the cohort was psychiatric disorder (40%), followed by hearing loss (32%), overexposure to noise (22%), ear surgery (7%), tube dysfunction (7%), idiopathic (6%), and others (head trauma, ototoxicity, otitis, rhinitis and sinusitis, barotrauma, cervical troubles, …).
To a reader, it sounds like these are the definite and verified causes of tinnitus in the sample. Would you mind revising how you use the word “etiology”?
The TRT guide does not consider the comorbid conditions.
Second: There are two main types of hearing loss: noise-induced hearing loss (NIHL) and age-related hearing loss (ARHL). Some patients can suffer of ARHL even though they were not overexposed to noise.
Third: the reasons for ear surgery were stapedectomy (4), tympanoplasty (2) and cholesteotoma removal (1).
Fourth: Reviewer #1 is comparing different figures. It is well-known that neither all individuals with HL have tinnitus nor all people with tinnitus have HL. In our cohort, 73% of subjects had hearing loss (HL), but only 32% of patients attributed their tinnitus to HL.
That is precisely the point. The Authors need to define the following:
- What is attributed as a cause of tinnitus by the patients?
- What are the medical facts?
Strong statements such as "…the predominant tinnitus etiology of the cohort was.." give the impression that the authors are talking about medical facts and not about patients' perceptions.
7) The TRT is a well described method (see Scherer, R.W., Formby, C., Gold, S. et al. The Tinnitus Retraining Therapy Trial (TRTT): study protocol for a randomized controlled trial. Trials 15, 396 (2014). https://doi.org/10.1186/1745-6215-15-396) and the name “TRT” should only be used when the protocol is fully followed. Based on the methodological description, it is apparent that the Authors modified the TRT method. Therefore, the modifications should be clearly stated, and the name “TRT” should not be used.
Response: We are not aware indeed of claiming the use of “TRT” to name our therapy. We explicitly state in lines 202-204 “This customized broadband sound stimulus was generated by using a custom-designed GUI (Figure 6), consisting of an enriched acoustic environment (EAE) of filtered broadband noise [24].” However, we compare our proposal with TRT throughout the manuscript, as it is certainly a kind of “modified TRT”.
The Authors should describe very precisely WHAT, HOW, FOR HOW LONG, and USING WHICH EQUIPMENT they do with the patients. Right now, the Authors talk more about what others did than they do about what THEY did:
The main goal of sound therapy is to vary the auditory processing at subconscious level aiming to stimulate habituation to the tinnitus perception [15]. Numerous sound stimuli have been proposed for tinnitus therapy in the past decades [22-23]. According to Jastreboff and Jastreboff, any type of sound which does not annoy, create discomfort or damage hearing, is better than silence [18]. However, Schaette and Kempter [24] used a computer model of the tinnitus-related hyperactivity to show that a personalized sound consisting of a broadband noise filtered by the audiometry of the patient should reduce tinnitus optimally. This customized broadband sound stimulus was generated by using a custom-designed GUI (Figure 6), consisting of an enriched acoustic environment (EAE) of filtered broadband noise [25]. As the HL curves of both ears are used for filtering the random noise, these stimuli are always stereo, regardless of the ear location of the tinnitus.
These customized EAE stimuli were given to the participants in mp3 format with instructions to hear them for one hour each day, over four months, using any audio player device (mp3 player, smartphone, computer …) connected to high quality headphones. Furthermore, they were trained in adjusting the stimuli volume to a level just below their tinnitus percept (the mixing point) [14]
Please reorganize that section and provide the technical data of the equipment used.
8) Although in the Methods, the Authors went into the trouble of sample stratification into with and without hearing loss, in the Results, the Authors do not demonstrate therapy outcome based on the patients’ hearing loss status. Why not?
Response: Yes, Reviewer #1 is right. We use stratification into HL and NH subtypes to provide some line of argument with an article recently published in Audiology Research (Reference [29]). However, we did not find differences in the therapy outcome of both patient subgroups.
If the Authors have not observed differences between the groups with and without hearing loss, they should state this. One sentence suffices.
